# Vulnerability of benthic trait diversity across the Mediterranean Sea following mass mortality events

J. Carlot [1] ✉, C. Galobart[2], D. Gómez-Gras[3,4], J. Santamaría [2], R. Golo[2], M. Sini [5], E. Cebrian[2], V. Gerovasileiou [6,7], M. Ponti [8,9], E. Turicchia [8,9], S. Comeau[1], G. Rilov [10,11], L. Tamburello[12,13], T. Pulido Mantas [14], C. Cerrano [14], J. B. Ledoux [15], J.-P. Gattuso [1,16], S. Ramirez-Calero[4,17], L. Millan [17], M. Montefalcone [18], S. Katsanevakis [5], N. Bensoussan [17], J. Garrabou[17] & N. Teixidó [1,19] ✉

Unraveling the functional future of marine ecosystems amid global change poses a pressing challenge. This is particularly critical in the Mediterranean Sea, which is highly impacted by global and local drivers. Utilizing extensive mass mortality events (MMEs) datasets spanning from 1986 to 2020 across the Mediterranean Sea, we investigated the trait vulnerability of benthic species that suffered from MMEs induced by nine distinct mortality drivers. By analyzing changes in ten ecological traits across 389 benthic species—constituting an extensive compendium of Mediterranean ecological traits to date—we identified 228 functional entities (FEs), defined as groups of species sharing the same trait values. Our findings indicate that of these 55 FEs were impacted by MMEs, accentuating a heightened vulnerability within specific trait categories. Notably, more than half of the mortality records showed severe impacts on calcifying and larger species with slower growth which mostly account for tree-like and massive forms. Altogether, we highlight that 29 FEs suffered extreme mortality, leading to a maximum increase of 19.1% of the global trait volume vulnerability over 35 years. We also reveal that 10.8% of the trait volume may have been temporarily lost over the last five years, emphasizing the risk of a rapid ecological transformation in the Mediterranean Sea.

There is compelling evidence that biodiversity loss may hinder ecosystem functioning, thereby jeopardizing nature's capacity to sustain human well-being[1–3]. This critical relationship between biodiversity and ecosystem functioning can be investigated using species traits, which are measurable organismal characteristics (e.g., maximum size, growth rates)[4,5]. Species traits directly influence how organisms respond to environmental changes and determine their contributions to ecosystem dynamics, profoundly shaping ecosystem responses to shifts in community composition[6]. For example, as marine heatwaves (MHWs) become increasingly common and severe due to rising ocean temperature[7,8], species displaying a more advantageous set of traits (e.g., motile with a high thermotolerance) can better withstand these rapid changes[9]. Consequently, species' ecological strategies resulting from beneficial trait combinations will significantly influence the future composition and functioning of ecosystems[10]. Therefore, assessing species trait diversity may provide critical insights into community dynamics and potential ecosystem functional changes driven by global and local mortality drivers[11,12]. Yet, trait-based approaches using long-term ecological data to quantify the capacity of trait diversity to buffer the effects of

global change on ecosystem functioning remain scarce, particularly within the marine realm[13].

Additionally, species loss may result in an increasing functional homogenization of biodiversity, characterized by the loss of the rarest and most vulnerable traits[14,15]. Such functional homogenization has the potential to modify the spectrum of ecosystem functions and affect the capacity of ecosystems to deliver goods and services[16]. For instance, despite the recovery of coral cover following mortality events, many tropical coral reefs worldwide exhibit a significantly reduced functional trait diversity compared to pre-disturbance levels[17,18]. Similarly, in temperate coastal ecosystems, the decline of seagrasses[19] and kelps[20] leads to their substitution by less diverse and productive assemblages, such as sea urchin barrens[21] or algal turfs[22]. This results in a decrease in habitat complexity with profound impacts on associated communities[23] and a shift in ecosystem functioning. Among marine ecosystems, the Mediterranean Sea stands out as particularly vulnerable, experiencing rapid warming and intensifying MHWs[8], along with numerous local mortality drivers (e.g., pollution, nutrient loadings, and diseases) that particularly affect key coastal species, such as seagrasses, canopy-forming seaweeds, and macroinvertebrates.

Here, we investigate the vulnerability of benthic trait diversity due to global and local mortality drivers, utilizing an unparalleled dataset of mass mortality events (MMEs) across the Mediterranean Sea, encompassing over 1858 mortality records from 1986 to 2020. Leveraging several databases, we assembled a comprehensive dataset of Mediterranean benthic species' traits, characterizing 389 species, based on ten ecological traits, and resulting in 228 functional entities (FEs, i.e., species with the same combination of traits). Specifically, we identified which FEs are the most impacted and quantified the mortality drivers' contribution to their impact over time and across regions. Our findings reveal that 10.8% of the trait volume has been impacted by global and local mortality drivers in the last 5 years, supporting the ongoing and rapid transformation of Mediterranean marine ecosystems[12].

## Results

We gathered mortality data related to MMEs in the Mediterranean Sea using existing datasets from the T-MEDNet platform (https://t-mednet.org/mass-mortality/mass-mortality-events) and a previous study which included data from 2015 to 2019[8] that we complemented with a literature review (see "Methods") in order to synthesize a comprehensive dataset on MMEs (Fig. 1a). We categorized MMEs into three severity levels: severe (mortality rate of colonies (colonial species) and individuals (non-colonial species) across sites and over time exceeding 60%), moderate (mortality rate of colonies and individuals across sites and over time between 30% and 60%), and low (mortality rate of colonies and individuals across sites and over time below 30%). Severe mortality accounts for over half of the observations (54%), while moderate and low mortality represent 20% and 26% of the events, respectively. Additionally, MMEs have been observed with increasing frequency, from 88 observations between 1986 and 1990 to 997 observations between 2016 and 2020. This increase in the frequency of MME observations parallels the rise in the intensity of mortality drivers, as suggested by the increased average Sea Surface Temperature (SST) serving as a proxy for temperature anomaly, average wind gusts higher than $15\,m\,s^{-1}$ serving as a proxy for storm occurrence, and average nutrient runoff serving as a proxy for water turbidity (Fig. 1b–d).

Subsequently, we assigned ten ecological traits to each benthic species and employed linear Bayesian models to predict their mortality severity. Our analysis revealed that mortality was more pronounced in specific ecological trait categories (Fig. 2), emphasizing the vulnerability of certain trait categories. Specifically, in 7 out of 10 ecological traits, a predominant trait category experienced

significantly higher mortality, surpassing other categories by a minimum of 5 to 8-fold. High mortality was most pronounced for long-living (>20 years), non-carbon storing, heterotroph, very large (>50 cm), slow-growing (ca. 1 cm/year), calcifying, and non-motile species, with values ranging from 72.6% to 83.0% (credible intervals [38.1%; 100%] to [55.1%; 100%]) (Fig. 2b, e–j). In contrast, two trait categories (instead of a single one) were the most impacted for feeding and morphological form traits. Active pumping-filter-feeders and passive filter-feeders, as well as tree-like and massive-erect species, were more sensitive than other trait categories, with average mortalities ranging from 47.8% to 83.5% (credible intervals [10.6%; 85.1%] to [55.0%; 100%]) (Fig. 2a, d). While non-colonial species displayed a slightly higher average mortality rate of 68.8% (credible intervals [30.0%; 100%]), there was no substantial difference between colonial/modular and non-colonial species (Fig. 2c).

To obtain a comprehensive overview of Mediterranean benthic assemblages (including species not reported as impacted during MMEs), we used additional trait databases from previous studies assessing traits of Mediterranean benthic communities (see "Methods"), which resulted in a total of 389 species, subsequently classified into 228 different FEs. Among these FEs, 68 represented between 2 and 12 species, while 160 FEs (i.e., 70.1%) represented only one species, highlighting their unique trait combination. We then defined the trait hypervolume through principal coordinates analysis (PCoA; Fig. 3a), ensuring optimal trait space quality by minimizing the median absolute deviation, resulting in a six-dimensional space. Subsequently, we quantified the decrease in the trait volume of Mediterranean benthic assemblages with the observed mortality severity (Fig. 3b). Our results demonstrate that the trait volume has been impacted up to a rate of 47.4% within 35 years, with lower rates associated with high damage severity. For the most severe mortality records (i.e., between 90% and 100% mortality), 29 FEs were impacted and the trait volume underwent a substantial reduction of 19.1%. More specifically, ochrophytes, echinoderms, and molluscs were the most severely impacted (mortality mostly exceeding 60%), followed by bryozoans, cnidarians, poriferans, and tracheophytes with moderate impact (mortality records ranging between 30% and 60%).

We further investigated the effect of local and global mortality drivers on the trait volume over the last 35 years. We categorized each global and local mortality driver as either abiotic (e.g., temperature anomaly, extreme storm) or biotic (e.g., disease, mucilage coverage). Nearly half of our dataset is associated with mortality induced by positive temperature anomalies, accounting for up to 85% of observed abiotic mortality drivers (Fig. 4a). Before the year 2000, most mortality events were caused by one or two drivers, with modest effects on overall trait diversity. Impacted trait volume varied between 0 to a maximum of 10.6%, with credible intervals up to [0%; 37.1%]. During the same period, biotic mortality drivers were mainly represented by diseases, which significantly impacted up to nine FEs within a year, twice more than any other biotic mortality drivers (Fig. 4b). Diseases impacted trait volumes from a maximum of 1.8% (a single FE impacted) to a maximum of 9.1% (a maximum of nine FEs impacted) with credible intervals ranging from [0%; 9.0%] to [0%; 83.7%].

In the last decade, we observed a substantial rise in the occurrence of mortality events involving more than three mortality drivers within the same year, including temperature anomaly, storm, disease, and mucilage coverage. This upsurge, in both frequency and types of mortality drivers, led to a remarkable increase in traits impacted by mortality, with up to 26.3% of the total trait volume impacted in 2019 (credible intervals [2.5%; 54.5%]) (Fig. 4c). This sharp increase in impacted trait volume underlines higher vulnerability in the trait diversity over time. We then averaged the impacted trait volume by decade and quantified trait vulnerability up to 0.6% in the 1990s (with an average of 3.1 ± 5.3 FEs impacted) compared to a maximum of 7.1% in the 2010s (with an average of 12.5 ± 11.3 FEs impacted). This decline

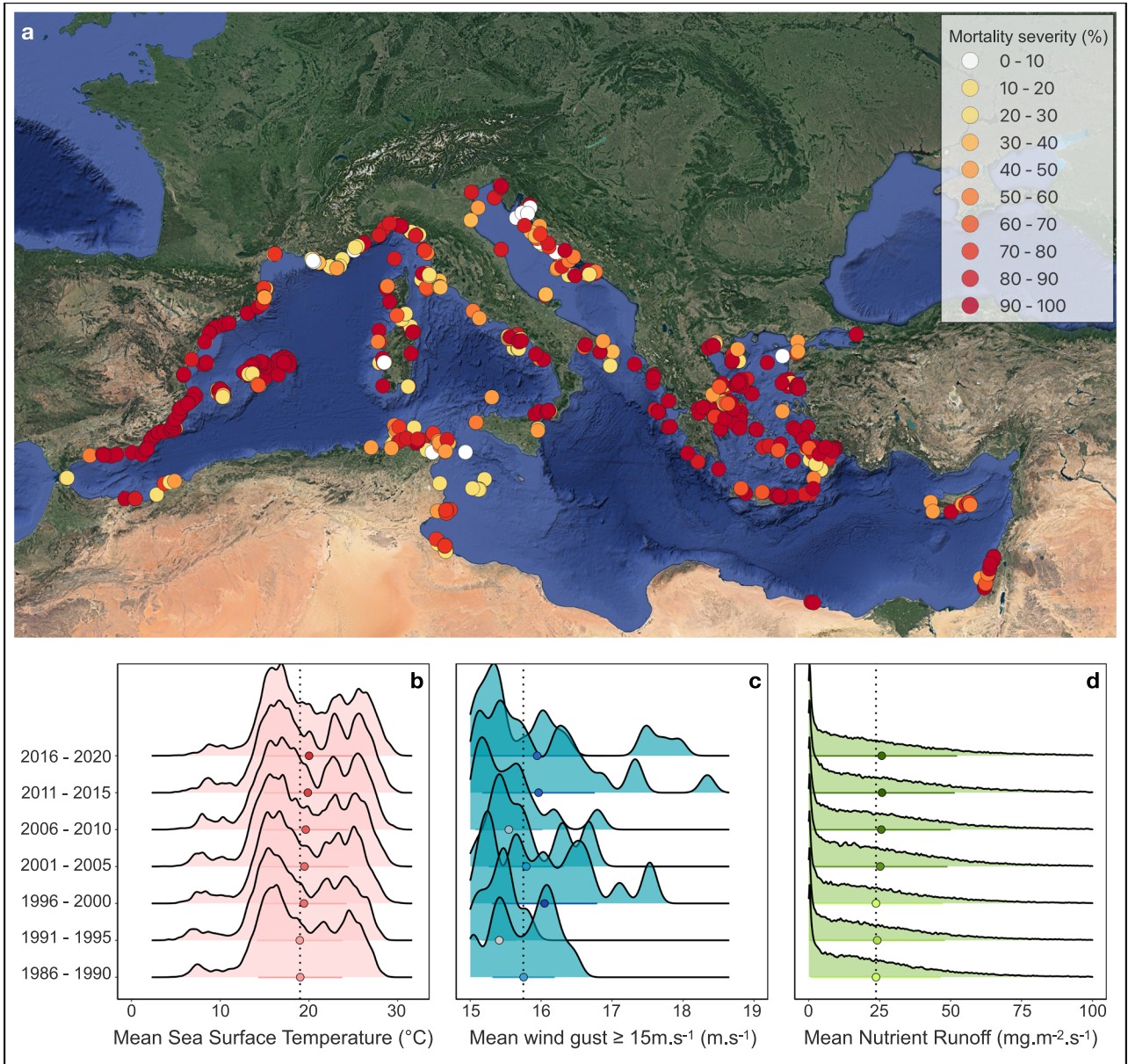

**Fig. 1 | Distribution of mass mortality events (MMEs) across the Mediterranean Sea. a** Spatial representation of 1858 mortality records according to damage severity from 1986 to 2020. Each data point represents a quantified MME in a single species population, related to a possible biotic or abiotic mortality driver at a distinct georeferenced site and year. **b–d** The ridgeline plots display the distribution of three proxies for three of the main mortality drivers studied over the last 35 years. Mean sea surface temperature, mean wind gusts higher than 15 m s⁻¹, and mean nutrient runoff have been quantified using Copernicus data (https://climate.copernicus.eu) across the Mediterranean Sea for a range of 5 years. These three metrics serve as proxies for temperature anomaly, storms, and turbidity.

in trait volume is even more accentuated if we average by 5 years, with a notable maximum of 10.8% impact across the Mediterranean basin in the most recent years. To ensure that our findings were not biased due to an unbalanced sampling effort, we randomly picked 100 observations from each consecutive decade and defined the corresponding trait volume, repeating this process 1000 times, and validating the escalating vulnerability observed in trait volume over time.

We finally investigated the extent of trait diversity impact across a longitudinal gradient (Fig. 5a) and unveiled differential impacts on trait diversity across three main regions (Western, Central, and Eastern). Most records (ca. 70%) were reported in the western region, with the remaining 30% evenly distributed between the central and eastern regions of the Mediterranean Sea (Fig. 5b). Then, to compare regions with the same sampling effort, we randomly picked 100 observations across the three main regions to define the trait volume and the

number of FEs, and repeated this process 1000 times to strengthen our analysis (see "Methods"). We observed a longitudinal gradient in the impacted trait volume: 14.3% ± 4.4% in the western region, 11.2% ± 3.9% in the central region, and 9.8% ± 4.1% in the eastern region. However, the number of FEs impacted was slightly higher in the central region with 21.9 ± 2.0 FEs, followed by 21.3 ± 2.4 FEs in the western region, and 17.5 ± 1.8 FEs in the eastern region (Fig. 5c), pointing to out that impacted FEs in the western region were more diverse than other regions. More precisely, no impact was observed for tracheophytes and echinoderms in the central region nor for tracheophytes, ochrophytes, and chordates in the eastern region. Finally, considering all the observations across regions, we also quantified that the maximum impacted trait volume occurred in the last 5 years, with the western region reaching values of 11.6% (in 2018), as opposed to 5.4% in the central region (in 2017) and 2.3% in the eastern region (in 2019) (Fig. 5 d).

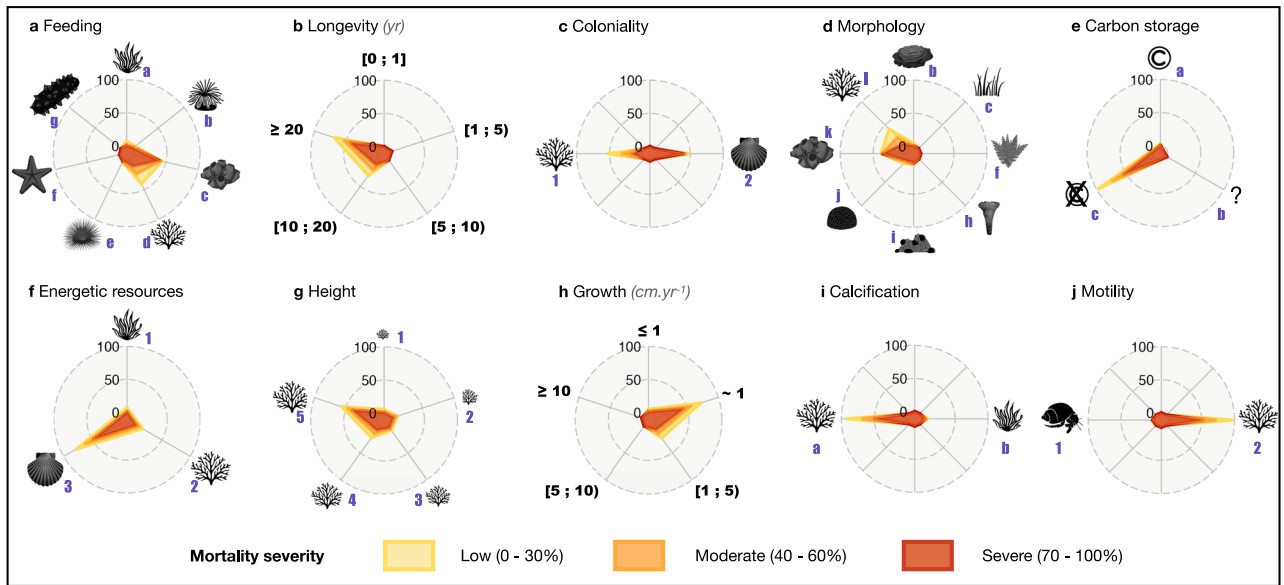

**Fig. 2 | Distribution of the number of observations according to the mortality severity (Low, Moderate, and Severe) on ten traits in the Mediterranean Sea.** **a** Feeding (a no, autotrophs, b. active filter feeders with cilia, c. active filter feeders by pumping, d. passive filter feeders, e. herbivores and grazers, f. carnivores, and g. detritivores), **b** Maximum longevity (1. lower than one year to one year, 2. two to five years, 3. five to ten years, 4. ten to twenty years, and 5. more than 20 years), **c** Coloniality (1. colonial/modular, 2. solitary), **d** Morphological form (b. encrusting, c. filamentous, f. articulated, h. cup-like, i. massive-encrusting, j. massive-hemi-spheric, k. massive-erect, and l. tree-like), **e** Carbon storage (a. yes, b. potentially, and c. no), **f** Energetic resources (1. photosynthetic autotroph, 2. photo-heterotroph, and 3. heterotroph), **g** Height (1. very small, 2. small, 3. medium, 4. large, and 5. very large), **h** Growth rate (1. extreme slow i.e., lower than 1 cm yr$^{-1}$, 2. slow i.e., ca. 1 cm yr$^{-1}$, 3. intermediate i.e., 1–5 cm yr$^{-1}$, 4. fast i.e., 5–10 cm yr$^{-1}$, and 5. very fast i.e., more than 10 cm yr$^{-1}$), **i** Calcification (a. with calcareous structures, b. without calcareous structures), and **j** Motility (1. sessile and 2. vagile). See Supplementary Table 3 for trait category descriptions. Radial marks at 0%, 50%, and 100% display the percentage of observations according to the trait category within the ecological trait due to mass mortality events (MMEs). The mortality severity ranges from: low (from 0% to 30%, in yellow), to moderate (from 40% to 60%, in orange), and severe (from 70% to 100%, in red).

## Discussion

By compiling a comprehensive marine temperate mortality database and their associated species-specific traits, we reveal a prevalent trend of increased ecological trait vulnerability in the last decade. Given the likely expansion of mass mortality events both spatially and temporally under global change, this trend may induce trait homogenization and reshape the ecological trait characteristics of benthic communities throughout the Mediterranean Sea. Over the 1858 mortality records observed from 1986 to 2020, more than half showed severe impacts, with recurrent effects on the same FEs. Specifically, our trait analysis identified a significantly higher vulnerability of a single trait category for each trait, surpassing other categories by a minimum of 5–8-fold. This concern of trait homogenization is emphasized by the fact that 10.8% of the trait volume may have been impacted over the past five years across the Mediterranean. Although caution is warranted, our results suggest an increase in the vulnerability of the overall trait diversity, threatening the assemblages of benthic communities in the Mediterranean Sea.

More precisely, the Mediterranean Sea is a global hotspot of local extinctions[24], and certain trait categories are consistently more impacted. Motile species exhibit lower impacts than sessile species, as they can find temporary refugia from short-term perturbations, such as MHWs[23], but might likely be impacted in the face of acute perturbations. Habitat-forming species, such as corals and gorgonians, present a narrow thermal range and face greater risk. The pointed-out vulnerability in calcifying and habitat-forming organisms could lead to the collapse of many food webs[25]. Moreover, higher mortality rates among larger species with extended lifespans and slower growth rates, particularly among heterotrophs, might hint at potential cascading effects, leading to community restructuring towards lower trophic levels, with profound consequences for Mediterranean biodiversity and ecosystem dynamics[26]. Furthermore, filter feeders, both active (by pumping) and passive, are disproportionately impacted, signaling a potential reduction in critical ecological functions, such as processing large volumes of water to capture particles[27], which could further affect nutrient cycling and carbon flows. Most impacted species exhibit a massive erect or tree-like morphology, further contributing to the potential decline in overall structural complexity[28,29]. Although some MMEs are more difficult to assess than others, these findings collectively suggest that the increasing mortality events on specific traits might yield important ecological shifts in the long term, with far-reaching implications. Hence, a more comprehensive examination of the intricate interplay between species traits, ecological functions, and the broader dynamics of Mediterranean marine ecosystems is urgently needed.

On the other hand, a prolonged high trait vulnerability may lead to an extensive loss of species with unique trait categories (e.g., tree-like species, mostly representing habitat-forming species) and may also pave the way for the establishment of new species (e.g., alien species). These new species may be better adapted to present and future local environmental conditions and display higher resistance and resilience to environmental perturbations due to certain traits[30]. Currently, such species are mostly coming from the Red Sea through the Suez Canal[31–33]. The disproportionate impact of invasive alien species on extant communities might result in a functional diversity shift, leading to a greater homogenization and simplification of the ecosystem[32]. In rare cases, it might also lead to a significant increase in trait diversity. For example, the native mollusk assemblages in the eastern Mediterranean shallow benthic environments have been almost completely replaced by tropical alien species[34] possessing distinct traits, thereby altering the overall ecosystem functioning[35]. However, if a prolonged high trait vulnerability may lead native species being replaced by alien species with similar traits, our models suggest that these species may be also vulnerable to the mortality drivers identified in this study and thus, highly threatened in the long term.

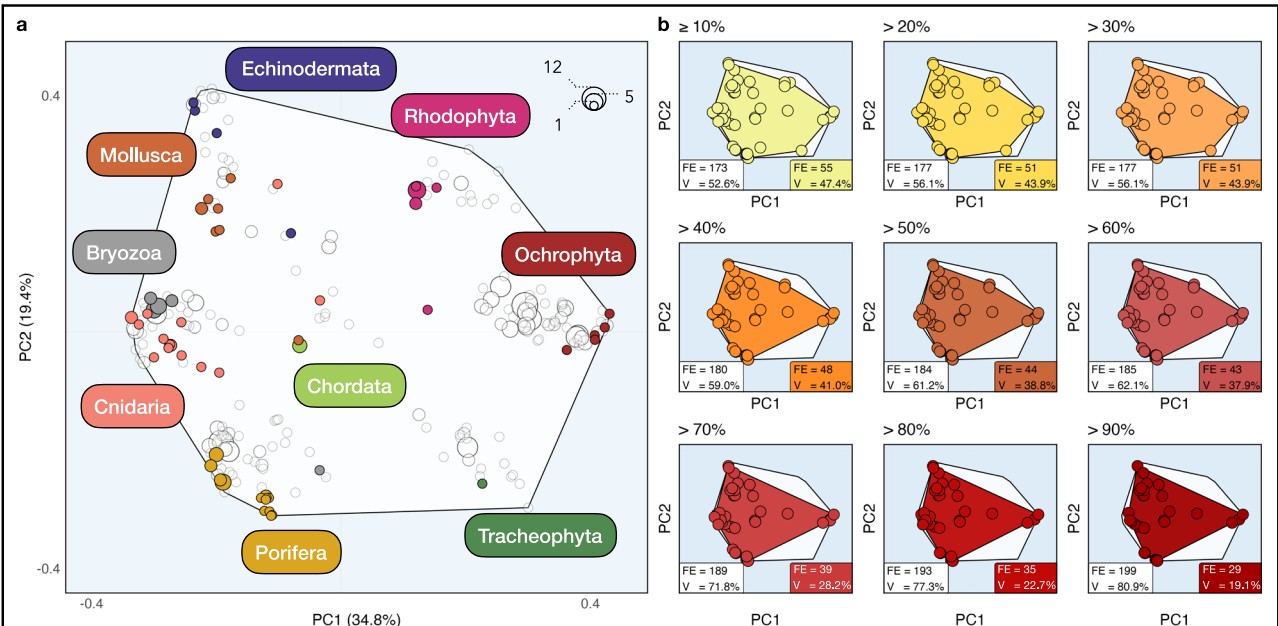

**Fig. 3 | Global trait space occupied by functional entities (FEs) of benthic species impacted and non-impacted by mortality drivers across the Mediterranean Sea.** The trait space for the global pool of 389 benthic species and 228 FEs. The axes (PCoA1 and PCoA2) represent the first two dimensions of the 6D functional space collectively explaining 54.2% of the total deviance. **a** Global trait space of benthic Mediterranean species. The FEs are non-impacted are filled in white, whereas the FEs damaged by mortality drivers are filled in color. The size of the dots represents the number of species within a FE. **b** Trait space of FEs impacted by mortality drivers following mortality severity: from ≥10% to >90% damaged. The legends within each panel show the number of non-impacted FEs (in white) and damaged FEs (in color) and the corresponding trait volume (V). The global trait space is filled in white.

Over the past four decades, the Mediterranean Sea has faced an increase in both frequency and intensity of mortality drivers, making it one of the most impacted regions globally[36,37]. Nevertheless, it is important to highlight that MMEs resulting from these mortality drivers are quite diverse. Necrosis observed on gorgonians does not necessarily imply the mortality of the colony as it may recover over time. This is in contrast to mortality from disease, such as that observed with Pinna nobilis. However, if these mortality drivers persist or increase over time, MMEs, despite their diverse nature, will likely lead to population declines or local extinctions[24], thereby increasing the loss of the associated trait categories. Our results support this increase in frequency and types of mortality drivers over the years. Trait diversity has consequently been impacted with increasing frequency by biotic mortality drivers such as diseases or predator outbreaks[38], and by abiotic mortality drivers including pollution, water turbidity due to enhanced sediment loads, eutrophication, or anoxia[39]. Furthermore, the effects of global change have become increasingly pronounced in this region, as seawater temperature has increased steadily since the 1970s[40]. This trend has been further underscored by the heightened frequency and intensity of MHWs in recent decades[8,41].

Our results demonstrate that mortality drivers are not homogeneously distributed within the Mediterranean Sea. The western region seems to be more vulnerable, followed by the central and the eastern regions, respectively. The reason for the comparatively lower number of mortality drivers and impacted trait diversity in the central and eastern regions may also be due to a research effort bias rather than a lower number of existing pressures. Several parts of the eastern Mediterranean basin have been severely impacted by MHWs and the expansion of invasive alien species over the past decades. While previous biodiversity studies have shown a decline in the richness of native species from the northwestern to the southeastern parts of the Mediterranean[36,42], the opposite trend has been observed for the diversity of invasive alien species and the rise in seawater temperature[43]. These mortality drivers, acting in concert with other human-induced pressures, have altered the

seascape of these areas through the increase of rocky barrens[33,42] and the local extinction of native species[43], which have led to a profound restructuring of marine communities with unknown consequences. The overall lack of time series on past community composition in the central and eastern basin[44,45] hinders the quantification of the magnitude of the ecological shifts[46], except perhaps for a few cases that have revealed dramatic changes in coastal rocky habitats[47]. Hence, we call for caution regarding any longitudinal gradient observed as it is challenging to unravel the cause of the lower number of mortality records and a number of impacts in these areas, which could stem either from the fact that such MMEs remained unnoticed in the past or from a higher increase in trait loss from west to east.

Theoretically, ecosystems could mitigate the risk of trait diversity reduction driven by species losses if both trait redundancy and response diversity coexist within the system. That would be the case, if the functions that are being impacted by the loss of a given species could be sustained by at least one other tolerant species that is functionally redundant (i.e., sharing a similar combination of traits)[48]. However, far from this ideal theoretical context, empirical evidence increasingly suggests that many tropical and temperate marine ecosystems such as coral reefs and kelp forests are experiencing decays of functionally unique or irreplaceable species (e.g., foundation species) due to an increase in the vulnerability of trait diversity which leads to shifts in functionality and loss of associated ecosystem services[18,28,49]. Here, we contribute to this growing body of evidence on trait vulnerability increase by revealing that, at a scale never addressed before in the Mediterranean, critical trait categories (e.g., tree-like or long-lived species) are also being significantly impacted across the entire basin. This process is simultaneously impacting multiple benthic ecosystems and a vast array of unique Mediterranean traits associated with many endemic species. Continued MMEs linked to a maintained or increased vulnerability of trait diversity such as the one observed in this study could eventually harm the functional resilience of Mediterranean benthic ecosystems in the future.

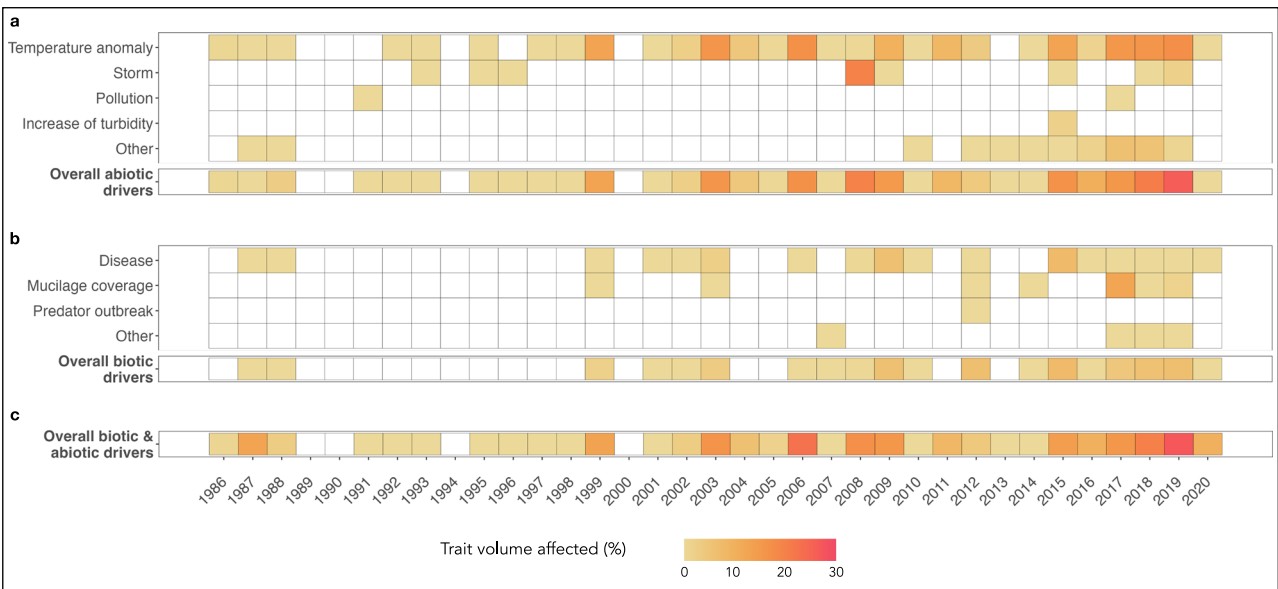

**Fig. 4 | Impact of global and local mortality drivers on trait volume across the Mediterranean Sea from 1986 to 2020.** The heatmap represents the volume of the trait space impacted by mortality drivers over time. **a** Abiotic mortality drivers are: temperature anomaly, storm, pollution, an increase of turbidity, others, and their combination; **b** biotic mortality drivers are: disease, mucilage coverage, predator outbreak, others, and their combination; and **c** combination of both biotic and abiotic mortality drivers on the ecological trait volume. The trait volume space has been calculated for each mortality driver, as well as for overall mortality drivers over time and color-coded accordingly.

Overall, our results point to a concerning trajectory for Mediterranean Sea benthic ecosystems amid global change. The consistent impact on recurrent ecological trait categories not only signals a potential increase in the vulnerability of the overall trait diversity over time, but also raises concerns about the future potential loss of unique trait categories and associated ecosystem functions. While the introduction of alien species sharing similar traits is likely to arise, their survival may also be at risk in the face of multiple mortality drivers in the coming years. Indeed, as ongoing pressures are predicted to intensify[50], changes in benthic composition within the Mediterranean Sea are very likely. These changes might either simplify the system, with a reduced trait diversity, or lead to a novel system with the potential emergence of new traits[35] with unknown consequences for ecosystem functions and related services in the future.

## Methods

### Data collection for mortality records
Mortality data have been acquired from three main sources: (1) from the T-MEDNet platform collecting data since 2012 (https://t-mednet.org/mass-mortality/mass-mortality-events)[51] (hereafter, T-MEDNet dataset); (2) from a dataset which include data from 2015 to 2019[8] (hereafter, mortality dataset 2015–2019); and (3) a literature review (see Supplementary Table 1 for details of the datasets used). The T-MEDNet dataset contains mortality data assessed and validated exclusively by scientific experts, accounting for 710 records across a wide spectrum of benthic species. The mortality dataset 2015–2019[8] was obtained through a collaborative effort across the Mediterranean Sea and represents a comprehensive inventory of MME records for benthic species in the region from 2015 to 2019, accounting for 1125 records. Finally, for this study, we performed a literature review based on the Scopus and the Web of Science Core Collection databases, carried out on 16th November 2021. Manuscripts were retrieved by applying the following research query: (mortalit* OR necros* OR diseas* OR bleaching OR heatwave* OR heat wave*) AND (marine OR sea OR ocean OR seabed OR benth* OR reef*) AND (Mediterranean OR Adriatic OR Levant* OR Aegean OR Alboran OR Tyrrhenian OR Balearic OR Ligurian OR Ionian OR Sicily OR Tunisian OR Sidra OR Catalan) to the title, keywords and abstract fields. According to our inclusion criteria, only non-duplicated English manuscripts with empirical observations published between 1979 and 2020 were retained, leading to a total of 3536 studies on Scopus and 2895 studies on Web of Science to screen. Manuscripts from these two sources were merged, and duplicate entries were removed. The retained manuscripts ($n = 3860$) were closely inspected to identify possible mortality data not present in previous datasets, and any relevant data were extracted. At this stage, 588 manuscripts and their data were retained. These data sources were merged into a single dataset and further inspected for possible duplicate records (i.e., close geographic coordinates and dates) by mapping records in the QGIS cloud. Finally, records not focusing on benthic species (e.g., fishes) were disregarded. The literature review contributed data from 58 studies, providing 266 records, leading to an overall dataset of 2101 entries. Finally, for 243 entries, MMEs had been observed but not quantified, resulting in a final dataset of 1858 records useful for data analysis. Each entry is defined by a quantified MME on a single species population, related to a possible biotic or abiotic mortality driver at a distinct georeferenced site and year. Based on the mortality dataset 2015–2019[8], we considered impacted colonies or individuals displaying signs of recent mortality based on the following criteria: (1) denuded skeletons or tissue necrosis over 10% of its surface in gorgonians, sponges, and scleractinian corals and empty valves in bivalves attached to the substratum; and (2) increase of shoot mortality or sharp decline on thallus densities for seagrasses and habitat-forming macroalgal species.

### Database description of mortality records
The database contains information about mortality occurrences of benthic species. For each mortality occurrence, the following main information is included: (1) species name and phylum; (2) year of the observation; (3) longitude and latitude; (4) depth range; (5) type of habitat; (6) type of mortality drivers (abiotic and biotic, see below); (7) percentage of mortality of dead colonies across sites and over time (from 10% to 100%); and (8) qualitative percentage of mortality

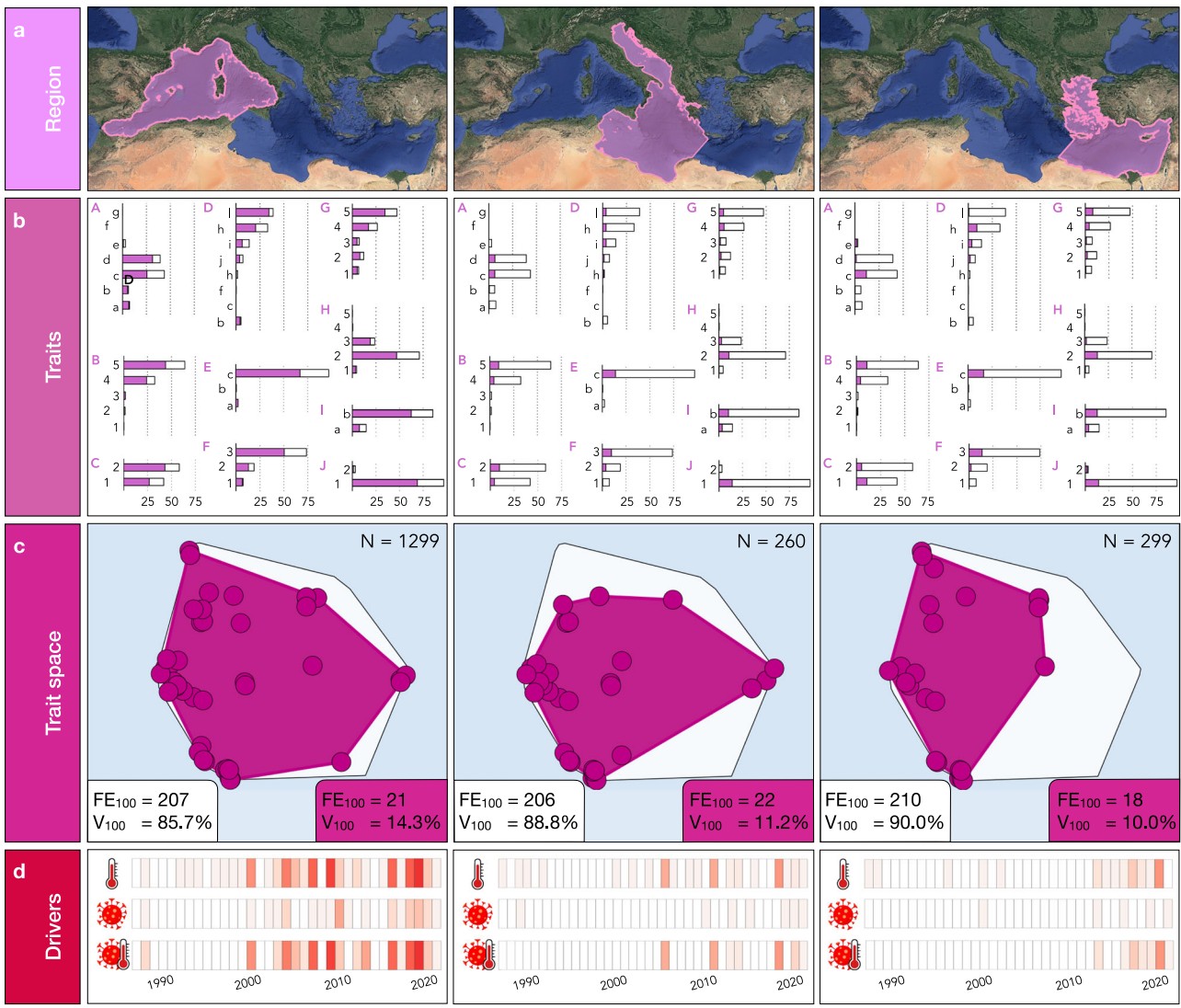

**Fig. 5 | Mortality trait patterns across three main regions (western, central, and eastern) in the Mediterranean Sea from 1986 to 2020. a** Spatial definition of the three main regions; **b** proportion of mortality records (solid) to total records (white) for each impacted trait at each region. Traits are A, Feeding (a. no, autotrophs, b. active filter feeders with cilia, c. active filter feeders by pumping, d. passive filter feeders, e. herbivores and grazers, f. carnivores, and g. detritivores), B, Maximum longevity (1. lower than one year to one year, 2. two to five years, 3. five to ten years, 4. ten to twenty years, and 5. more than 20 years), C, Coloniality (1. colonial/modular, 2. solitary), D, Morphological form (b. encrusting, c. filamentous, f. articulated, h. cup-like, i. massive-encrusting, j. massive-hemispheric, k. massive-erect, and l. tree-like), E, Carbon storage (a. yes, b. potentially, and c. no), F, Energetic resources (1. photosynthetic autotroph, 2. photo-heterotroph, and 3. heterotroph), G, Height (1. very small, 2. small, 3. medium, 4. large, and 5. very large), H,

Growth rate (1. extreme slow, 2. slow, 3. intermediate, 4. fast, and 5. very fast), I, Calcification (a. with calcareous structures and b. without calcareous structures), and J, Motility (1. sessile and 2. vagile). See Supplementary Table 3 for more information on trait category description; **c** trait space occupied by functional entities (FEs) in the three regions according to all records. The global trait space is filled in white. The total number of mortality records (N) is indicated on the top right of each panel. Due to heterogeneity in the number of observations across regions, the legends within each panel showing the non-impacted FEs (in white), damaged FEs (in color), and associated trait volume (V) display only 100 observations within the region; and **d** impact of abiotic, biotic, and combined mortality drivers on trait volume. Values ranged from 0 (white) to a maximum of 11.7% (red) in the Western basin in 2018.

grouped into three categories: low impact (mortality rate of colonies (colonial species) and individuals (non-colonial species) across sites and over time below 30%), moderate impact (mortality rate of colonies and individuals across sites and over time between 30% and 60%), and severe impacts (mortality rate of colonies and individuals across sites and over time exceeding 60%). For mortality drivers, we refer to variables representing drivers of mortality including abiotic (temperature anomalies, storms, pollution, and others) and biotic (predator outbreak, mucilage cover, disease, and others). In total, we had a dataset for 9 driver-mortality-related variables. See Supplementary Table 2 for a description of the database fields. In total, this study includes 1858 mortality records across 747 unique locations of 73 benthic species,

including 19 Porifera, 15 Cnidaria, 9 Rhodophyta, 9 Bryozoa, 8 Mollusca, 5 Ochrophyta, 4 Echinodermata, 3 Chordata, and 1 Tracheophyta.

## Trait characterization of benthic species

To obtain a comprehensive trait benthic dataset of the Mediterranean Sea, we combined four additional databases[28,29,52,53] of the benthic species observed in the Mediterranean Sea, totaling a total of 316 benthic species, in addition to the 73 benthic species impacted by MMEs assessed in this study (see Supplementary Table 1 for details of the datasets used). Altogether, we characterized the ecology of a total of 389 benthic species using ten ecological traits, which describe complementary facets of species

ecology. The traits were categorized as qualitative (defined for each following category with a lowercase letter) and semi-quantitative (defined for each following category with a number) and included (A) Feeding, consisting of seven categories: (a) no (autotroph), (b) active filter feeders with cilia, (c) active filter feeders by pumping, (d) passive filter feeders, (e) herbivores and/or grazers, (f) carnivores, and (g) detritivores; (B) Maximum longevity, consisting of five categories: (1) <1 year–1 year, (2) 2 years–5 years, (3) 5–10 years, (4) 10–20 years, and (5) ≥20 years; (C) Coloniality, consisting of two categories: (1) solitary and (2) colonial or modular or gregarious; (D) Morphological form, consisting of eight categories: (a) boring, (b) encrusting, (c) filamentous, (d) stolonial, (e) foliose-erect, (f) articulated, (g) coarse branched, (h) cup-like, (i) massive encrusting, (j) massive hemispheric, (k) massive-erect, and (l) tree-like; (E) Carbon storage, consisting of three categories: (a) yes, (b) potentially, and (c) no; (F) Energetic resource, consisting of three categories: (a) photosynthetic autotroph, (b) photo-heterotroph, and (c) heterotroph; (G) Height (size), consisting of five categories: (1) very small (<2 cm), (2) small (2–5 cm), (3) medium (5–20 cm), (4) large (20–50 cm), and (5) very large (>50 cm); (H) growth rates, consisting of five categories: (1) extreme slow (<1 cm yr$^{-1}$), (2) slow (ca. 1 cm yr$^{-1}$), (3) moderate (>1–5 cm yr$^{-1}$), (4) high (5–10 cm yr$^{-1}$), (5) very high (>10 cm yr$^{-1}$); (I) calcification, consisting of two categories, (a) without and (b) with calcareous structures; and (J) motility, consisting of two categories, (a) sessile and (b) vagile. Trait values were obtained and selected from the expertise of the team members. See Supplementary Table 3 for a detailed description of traits.

## Mortality severity across trait categories

To quantify mortality severity vulnerability across trait categories, we employed Bayesian linear regression (BLR) using the package *brms*[54], resulting in the development of ten models, each corresponding to a specific trait. Specifically, given that mortality occurrences are bound between 0 and 1, we utilized a zero-one inflated beta regression with Bayesian inferences:

$$P(Y = 0) = (1 - \omega - \zeta) + \omega B(\alpha, \beta) \tag{1}$$

$$P(Y = 1) = \zeta + (1 - \omega)B(\alpha, \beta) \tag{2}$$

Here, $\omega$ represents the probability of excess zeros, $\zeta$ is the probability of ones, and B ($\alpha$, $\beta$) denotes the beta distribution with shape parameters $\alpha$ and $\beta$. Precisely, the coefficients corresponding to trait categories ($\alpha$ and $\beta$) follow flat priors. The intercept in the zero-one inflated model follows a student-t distribution with degrees of freedom (3), location (0), and scale (2.5), allowing for some flexibility. The dispersion parameter $\phi$ follows a gamma distribution with shape parameters (0.01, 0.01). The parameter $\zeta$, is assigned a beta distribution with shape parameters (1, 1), while the parameter $\omega$ is assumed to be flat by default. We ran each BLR with two chains, 2000 draws per chain, and a warm-up period of 1000 steps, thus retaining 2000 draws to construct posterior distributions with credible intervals. Finally, we verified chain convergence with trace plots and confirmed that $R_{hat}$ (the potential scale-reduction factor) was equal to 1[55]. We then computed $R^2$ values for each regression to validate our models with values ranging from 0.54 to 0.72 (Supplementary Data 1).

## Measuring trait diversity: number of FEs and trait volume

Functional entities (FEs) were defined as groups of species sharing the same combination of trait values[56]. In total, the 389 benthic species were classified into 228 different FEs (mean = 1.71, max = 12 species per

FE). One hundred sixty FEs contained only one species (i.e., no functional redundancy). See Supplementary Data 2 for a detailed description of traits for each species. We further constructed a multidimensional trait space by assessing the values of the ten ecological traits across all FEs. We based our framework on previous research[57] and the R package *mFD*[58]. Specifically, we computed pairwise trait distances between species pairs based on the ten ecological traits utilizing the Gower metric. This metric accommodates the mixing of different variable types while assigning them equal weight. Subsequently, we performed principal coordinates analysis (PCoA) on the Gower dissimilarity matrix. To ensure the creation of an accurate trait space, we determined the number of PCoA axes based on the mean absolute deviation (mAD). The mAD was computed between the initial Gower distance among FEs (based on trait values) and the final Euclidean distance in the trait space. Optimal results were obtained with a trait space of six dimensions, yielding the lowest mAD value (0.055). This low value ensures that the 6D space faithfully represents trait-based differences between species. Convex hulls were computed within the same trait space to investigate the distribution of species according to their trait combinations. For each computation, we determined (i) the number of FEs present corresponding to the FE richness and (ii) the trait volume ($V$), calculated as the volume within the trait space that englobes all the FEs in the six-dimensional space.

Then, we first assessed both the trait volume and the extent of FEs impacted according to the degree of mortality. The degree of mortality was expressed quantitatively at intervals of 10% (from 10% to 100%) and qualitatively (i.e., low, moderate, and severe). Second, we defined the change in trait volume according to mortality drivers. We created different subsets based on the number of mortality drivers over 35 years totaling 420 datasets. Subsequently, we determined the number of impacted FEs and the number of non-impacted FEs (by subtracting the total amount of observed FEs from the number of impacted FEs). If a single species within the FE was impacted while others not, we considered the whole FE vulnerable to MMEs. Similarly, we calculated the impacted trait volume using the overall trait space defined below. Third, to measure the change in trait volume across regions and to consider the heterogeneity in the sampling effort, we performed random sampling through MME observations using 100 observations across three geographical zones (western, central, and eastern basins). We repeated this process 1000 times to ensure robust results and calculate for each iteration impacted trait volume.

## Relationship between trait volume and local and global mortality drivers

To strengthen our results, we further seek to build a relationship between the impacted trait volume over time and both local and global mortality drivers. Given that the volume is also bounded between 0 and 1, we applied a Bayesian generalized linear model (GLM) using a zero-one inflated beta distribution. In this case, the predictor variables represent the number of FEs impacted by the considered mortality drivers, and we formulated the model as follows:

$$P(Y = 0) = (1 - \omega - \zeta) + \omega B(\alpha_{driver\_1}, \beta_{driver\_1}, \ldots, \alpha_{driver\_n}, \beta_{driver\_n}) \tag{3}$$

$$P(Y = 1) = \zeta + (1 - \omega)B(\alpha_{driver\_1}, \beta_{driver\_1}, \ldots, \alpha_{driver\_n}, \beta_{driver\_n}) \tag{4}$$

Similar to the previous models, $\omega$ represents the probability of excess zeros, $\zeta$ is the probability of ones, and B($\alpha_{driver\_1}$, $\beta_{driver\_1}$, ..., $\alpha_{driver\_n}$, $\beta_{driver\_n}$) denotes the beta distribution with shape parameters ($\alpha_{driver\_1}$, ..., $\alpha_{driver\_n}$) and ($\beta_{driver\_1}$, ..., $\beta_{driver\_n}$). Consistently, we maintained non-informative priors, and the same prior information was retained. The model was run with two chains, each producing 3000 draws, and a warm-up period of 1000 steps. We retained 4000 draws

in total to construct posterior distributions with according to credible intervals. To ensure chain convergence, we examined trace plots and confirmed that $R_{hat}$ equaled 1[55]. Finally, we computed $R^2$ values to validate our model, achieving values of up to 0.75 (Supplementary Data 1). All of the statistical analyses were run with the statistical software R version 4.2.1.

## Reporting summary
Further information on research design is available in the Nature Portfolio Reporting Summary linked to this article.

## Data availability
Mortality data have been acquired from the T-MEDNet platform (https://t-mednet.org/mass-mortality/mass-mortality-events)[51], a dataset that includes data from 2015 to 2019[8]; and a literature review. This data has been compiled in a single document as "MME-Review data". Benthic trait data have been acquired from four additional databases[28,29,52,53]. This data has been compiled in a single document as "Complete_Traits". Data, including data to generate all figures, are available at https://github.com/JayCrlt/MMEs_Mortality/Data/R[59].

## Code availability
The code, including to generate all figures, is available at https://github.com/JayCrlt/MMEs_Mortality[59].

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

## Acknowledgements

We thank E. Ballesteros for helping in providing information on benthic organisms' traits. This research was supported by the French National Research Agency Investments for the Future "4Oceans-Make Our Planet Great Again" grant, ANR-17-MOPGA-0001, and the National Recovery and Resilience Plan (NRRP), Mission 4 Component 2 Investment 1.4—call for tender no. 3138 of 16 December 2021, rectified by decree no. 3175 of 18 December 2021 of the Italian Ministry of University and Research funded by the European Union—NextGenerationEU; award number: project code CN_00000033, concession decree no. 1034 of 17 June 2022 adopted by the Italian Ministry of University and Research, CUP C63C22000520001 Project title "National Biodiversity Future Center—NBFC". M.S. and S.K. were supported by the European Union's Horizon 2020 research and innovation program HORIZON-CL6-2021-BIODIV-01-04 under grant agreement no. 101059877 "GES4SEAS—achieving good environmental status for maintaining ecosystem SErvices, by ASsessing integrated impacts of cumulative pressures".

## Author contributions

Conceptualization: J.C. and N.T. Data curation (Literature review): Led by M.P., E.T., and J.G.; Contribution from: J.C., C.B., D.G.G., J.S., R.G., M.S., E.C., V.G., S.C., G.R., L.T., T.P.M., C.C., J.B.L., J.P.G., S.R.C., L.M., M.M., S.K., N.B., and N.T. Data curation (trait characterization): led by N.T.; contribution from C.G., D.G.G., J.S., R.G., E.C., L.T., M.S., and V.G. Formal analysis: J.C. Investigation: J.C., N.T., C.G., D.G.G., J.S., E.C., M.S., V.G., L.T., G.R., J.G., and S.C. Writing (first draft): led by J.C.; contribution from N.T., S.C., C.G., J.S., D.G.G., L.T., M.S., and V.G. Writing (review and editing): led by J.C.; contribution from C.B., D.G.G., J.S., R.G., M.S., E.C., V.G., M.P., E.T., S.C., G.R., L.T., T.P.M., C.C., J.B.L., J.P.G., S.R.C., L.M., M.M., S.K., N.B., J.G., and N.T. Funding: N.T.

## Competing interests

The authors declare no competing interests.

## Additional information

[1]Laboratoire d'Océanographie de Villefranche, Sorbonne Université, CNRS, Villefranche-sur-mer, France. [2]Centre d'Estudis Avançats de Blanes (CEAB-CSIC), Girona, Spain. [3]Hawai'i Institute of Marine Biology, University of Hawai'i at Mānoa, Kaneohe, HI, USA. [4]Facultat de Biologia, Departament de Biologia Evolutiva, Ecologia i Ciències Ambientals, Institut de Recerca de la Biodiversitat (IRBIO), Universitat de Barcelona, Barcelona, Spain. [5]Department of Marine Sciences, University of the Aegean, University Hill, Mytilene, Lesvos Island, Greece. [6]Faculty of Environment, Department of Environment, Ionian University, Zakynthos, Greece. [7]Institute of Marine Biology, Biotechnology and Aquaculture (IMBBC), Hellenic Centre for Marine Research (HCMR), Heraklion, Greece. [8]Department of Biological, Geological and Environmental Sciences, University of Bologna, Ravenna, Italy. [9]Consorzio Nazionale Interuniversitario per le Scienze del Mare, Rome, Italy. [10]National Institute of Oceanography, Israel Oceanographic and Limnological Research (IOLR), Haifa, Israel. [11]Department of Marine Biology, The Leon H. Charney School of Marine Sciences, University of Haifa, Haifa, Israel. [12]Department of Integrative Marine Ecology, Sicily, Stazione Zoologica Anton Dohrn, Palermo, Italy. [13]National Biodiversity Future Center (NBFC), Palermo, Italy. [14]Department of Life and Environmental Sciences, Polytechnic University of Marche, Ancona, Italy. [15]CIIMAR, Matosinhos, Portugal. [16]Institute for Sustainable Development and International Relations, Paris, France. [17]Institute of Marine Sciences-CSIC (ICM-CSIC), Barcelona, Spain. [18]Department of Earth, Environment, and Life Sciences (DISTAV), University of Genoa, Genoa, Italy. [19]Stazione Zoologica Anton Dohrn, National Institute of Marine Biology, Ecology and Biotechnology, Ischia Marine Center, Ischia, Italy.
✉e-mail: jay.crlt02@gmail.com; nuria.teixido@szn.it

