## [Peer Review file · Nature Communications]

Vulnerability of benthic trait diversity across the Mediterranean Sea following Mass Mortality Events

Corresponding Author: Dr Jérémy Carlot

Please note that the reviewer numbering changes between rounds. Reviewer #1 in the first round is Reviewer #3 in the second round, Reviewer #2 in the first round is Reviewer #1, and Reviewer #3 is Reviewer #2 in the second round.

Version 0:

Reviewer comments:

Reviewer #1

(Remarks to the Author)

I was asked to review this manuscript having already reviewed it during a previous submission to another journal. In offering comments on this piece of work, I have therefore attempted to assess both the quality and clarity of the manuscript as an independent piece of work, but also the extent to which the authors have addressed the feedback provided by myself and other reviewers during the previous submission.

The manuscript presents a thorough and detailed analysis of the susceptibility of different functional entities (i.e., trait combinations) to biotic and abiotic drivers of mass mortality across benthic assemblages in the Mediterranean Sea. The analysis focuses on a database of mass mortality events acquired from various sources following an extensive compilation process and review of existing literature. Again, as I mentioned in my previous review, it is clear from reading the manuscript that considerable effort has gone into both compiling the database and carrying out the subsequent analyses of the resultant dataset. The results of this work offer valuable insight into the relative impacts of different mortality drivers on particular trait combinations and highlight spatial signals in the differing vulnerabilities of assemblages across the Mediterranean region. The manuscript also demonstrates an increase in the recurrence of mass mortality events in the Mediterranean, which, whilst not necessarily novel insight, is significant as it confirms localised observations are also persisting at regional scales. However, having had the opportunity to re-read the manuscript I am not wholly convinced that the authors have fully addressed the concerns raised during the previous review. I emphasise, however, that I still feel this is remediable.

Concerns raised during the previous review largely centred around the fact that the article was overreaching with its inference of having demonstrated 'trait erosion', and that there was insufficient detail provided regarding exactly what data was being used for the analyses. To address the first of these concerns the authors have replaced any mentions of 'trait erosion' with terminology relating to 'trait vulnerability' and infer that their data shows 'trait impairment' following mass mortality events. To me the use of 'trait vulnerability' fits with the fact that authors are showing the extent to which mass mortality events impacted upon particular trait combinations. However, the use of the term 'impairment' still implies a temporal aspect to the data, and the monitoring of trajectories following impact, that does not exist within the data as described. All the data shows is the instantaneous impact of observed mass mortality events on different functional trait combinations. Reading the manuscript it also felt that there were many instances where 'trait erosion' had been replaced with 'trait vulnerability' or 'impairment' without also addressing the context of the associated passage of text; meaning the passage no longer makes sense. This is particularly true of Lines 207-209 'Although caution is warranted, our results suggest an increase in the vulnerability of the overall trait diversity, especially if replacement by species with similar traits does not occur'. The over-inference of the results is also highlighted across Lines 440-442, in which the authors state that 'If a single species within [a functional entity] was impaired while others [where] not, we considered the whole [functional entity] vulnerable to [the mass mortality event]'. Whilst this is a perfectly acceptable, and articulated, assumption, it does represent an over estimation of vulnerability; the implications of which are currently not discussed further within the manuscript. In summary of this point, I still feel the inferences being made by the article need toning down.

Finally, although the data used is now better described in the methods, this same clarity is not reflected across the rest of the manuscript. This is not a trivial issue, since in Nature Communications the methods section comes at the end of the article,

and so key methodology details need to be articulated within the main text to provide context to the results. This issue is also key because the authors frequently state that the dataset that they are presenting, and using, represents a key advancement and component of the study – yet they still don't expand on what this dataset is.

I also detail a list of more minor concerns below:

Abstract:

Line 49: consider changing to 'we investigated the trait patterns'.

Line 54: consider removing 'on'.

Line 55-57: This feels like an odd example to use here, as surely the vulnerability of 'tree-like' morphologies centres more on the fact that Mediterranean octocoral species, which constitute the majority of tree-like morphologies, are physiological more susceptible to the mortality drivers, not that just because they are 'tree-like'?

Line 57-58: This statement is still inferring a temporal aspect to your data and assessment.

Line 59: Is impacted a better term than impaired? This point is relevant across the manuscript.

Introduction:

Line 68: consider removing 'the'.

Line 69: responses instead of response?

Line 95: What constitutes a mortality record?

Results:

Lines 107-112: Do you mean colonies here, or individuals? Also applies to the passage of the methods section where these same criteria are outlined (Line 353).

Line 147: 'dwindling' is still a temporal inference.

Line 168: trait instead of traits?

Discussion:

Line 203-205: It is not clear what you are trying to say with this statement: 'Specifically, our trait analysis identified a significantly higher vulnerability of a single trait category for each trait, surpassing other categories by 5 to 8-fold'?

Lines 230-243: This whole paragraph is discussing the implications of species loss, which is not being shown by the study.

Line 276: is 'raised' the correct verb and tense here?

Line 299: It is not clear what is meant by a 'potential trend towards trait diversity vulnerability'? This is an example of when it feels like you have replaced 'trait erosion' with 'trait vulnerability' without considering the wider context of the passage of text (see main concerns).

Methods:

Section 4.3 – Is there a clearer way to distinguish the different listing notation used across categories within traits, and the different traits used? Currently, the traits are listed using capitalised letters, whilst the within trait categories are listed using either lower case lettering or numbering. Overall, this makes the paragraph and listing difficult to follow.

(Remarks on code availability)

There is no code repository at the link provided.

Reviewer #2

(Remarks to the Author)

This is the second time I have seen the manuscript. The authors have made generally satisfactory responses to my concerns, and have improved clarity in general. I do have a few remaining substantial concerns:

* Data availability: the authors wrote in their response they would provide machine-readable open data. I didn't see it on the journal website yet; there is Table S5 in 527652_0_supp_9383202_sh84yj.pdf but this is not in CSV or similar format, and it is unclear if this data alone can be used to reconstruct the full study

* Code availability: the provided github: https://github.com/JayCrlt/MMEs_Mortality does not work when I tried to load it. I was unable to verify the code base.

Also a few minor concerns:

* many of the selected verbs still feel to me inappropriate in tone (e.g. 'unveiling', 'accentuating', 'weakening'). The authors may want to consider changing some of these, but it is really a stylistic choice.

* Fig 4: the legend is still not clear - do the colors represent the combined predicted effects of the listed variables, from a model? If so, describe the model in a bit more detail in the caption.

Benjamin Blonder

(Remarks on code availability)

See above.

Reviewer #3

(Remarks to the Author)

While the ms tries to address many of my points raised, it has not eliminated my concerns that it lacks scientific rigour. One important point remains, the potential collinearity between the sampling effort and the number of MMEs through time and external drivers (see below). The other the selection of traits and subset of fauna sampled which will drive the analysis towards specific traits sets to be identified as vulnerable (i.e. sessile large species that can easily be observed vs mobile species). And last the reduction of trait volume is a somewhat abstract measure, that appears attractive when telling the story, but is difficult to translate into real world scenarios. Overall I feel the paper is aiming very high, decorated with nice looking figures, but lacking rigour and substance. I do think its worthwhile publishing but not without an in depth exploration of its limitations (and as such I am not sure this is the right paper for nature communications).

I am still sceptical about some of the presented results especially with respect to the temporal trends. I think the authors should provide more details about the sampling effort through time. MMEs observation and sampling effort may have increased simultaneously with time, potentially confounding the data presented and the conclusions drawn. The authors should be transparent about their sampling effort and should provide measures to adjust for this imbalance in sampling, in their analysis and interpretation of the data. In the response to the reviewers comments this point has not yet been properly addressed.

"We also have to notice that although a consistent sampling effort over space and time is the ideal situation, it is impossible to be obtained on a large scale such as the entire Mediterranean basin involving 22 coastal countries and spanning 35 years. Still, the use of the best available datasets over large spatial and temporal scales can provide extremely useful insights and is a common practice in global or large regional assessments in ecological studies".

At least this point needs to be acknowledged and the limitations with respect to drawing firm conclusions need to be modulated.

The title should contain also the word benthic I suggest the following title: Trait diversity vulnerability across the Mediterranean Sea following Mass Mortality Events"

(Remarks on code availability)

Version 1:

Reviewer comments:

Reviewer #1

(Remarks to the Author)

This is the third time I have seen the manuscript. The authors have addressed my concerns about data availability. I successfully viewed the repository and confirmed the data / scripts in several files.

I thought the other reviewers raised important points and feel satisfied by the ways the authors have addressed them. I have just a few last language suggestions:

abstract: "we highlight that 29 FEs suffered extreme mortality, leading to a maximum of 19.1% of the global trait volume vulnerability over 35 years." - grammar is not clear to me, is this 'maximum loss'? something is missing here. also, 'may have been impacted' - how has it been impacted? 'temporarily lost'?

Figure S4, the new rarefaction analysis, is not cited in the main text.

(Remarks on code availability)

This looks good to go now.

Reviewer #3

(Remarks to the Author)

This is the third opportunity I have had to review this piece of work, and I feel that Carlot et al. have appropriately addressed my concerns. This piece of research comprises a huge amount of effort in synthesising an extensive database on the severity of mass mortality events across the Mediterranean and their impact upon functional trait entities. The manuscript now outlines the work with sufficient clarity as for a broad audience to follow and encompasses an insightful and thought-provoking discussion of the results. Thank you for the opportunity to review this piece of work.

I include below a few minor comments:

Line 54: consider changing to: 'indicate that 55 **of these** FE's were impacted by'.

Line 58: should this line be: 'leading to a maximum **increase** of 19.1%'?

Line 149 – 150: can you clarify what you mean exactly by 'with lower rates according to the damage severity'?

Line 180: consider changing to: 'picked 100 observations **from** each consecutive decade'.

Line 205: consider changing to: 'and **their associated** species-specific traits'.

Line 222: consider changing to: '**This** vulnerability **in** calcifying and habitat-forming organisms'.

Line 248: consider removing the word 'even'.

Line 321: The citation for the T-MEDNet platform should appear when this database is first mentioned on line 108.

(Remarks on code availability)

Exceptionally clear landing page to walk the readers through reproducing the analysis and figures. Nice use of a run-script.

REVIEWER COMMENTS

Reviewer #1 (Remarks to the Author):

I was asked to review this manuscript having already reviewed it during a previous submission to another journal. In offering comments on this piece of work, I have therefore attempted to assess both the quality and clarity of the manuscript as an independent piece of work, but also the extent to which the authors have addressed the feedback provided by myself and other reviewers during the previous submission. The manuscript presents a thorough and detailed analysis of the susceptibility of different functional entities (i.e., trait combinations) to biotic and abiotic drivers of mass mortality across benthic assemblages in the Mediterranean Sea. The analysis focuses on a database of mass mortality events acquired from various sources following an extensive compilation process and review of existing literature. Again, as I mentioned in my previous review, it is clear from reading the manuscript that considerable effort has gone into both compiling the database and carrying out the subsequent analyses of the resultant dataset. The results of this work offer valuable insight into the relative impacts of different mortality drivers on particular trait combinations and highlight spatial signals in the differing vulnerabilities of assemblages across the Mediterranean region. The manuscript also demonstrates an increase in the recurrence of mass mortality events in the Mediterranean, which, whilst not necessarily novel insight, is significant as it confirms localised observations are also persisting at regional scales. However, having had the opportunity to re-read the manuscript I am not wholly convinced that the authors have fully addressed the concerns raised during the previous review. I emphasise, however, that I still feel this is remediable.

We thank Reviewer 1 for their thorough analysis of this study and thoughtful feedback. We appreciate their recognition of the effort in compiling the database and conducting the subsequent analyses. We also value their optimism that the remaining concerns from the previous revision are remediable. In response, we have carefully revisited the main concerns regarding terminology (trait erosion, impairment) and their ecological implications, towing down when discussing overestimation of trait vulnerability, and better description of the methods section. See the following points below.

Concerns raised during the previous review largely centred around the fact that the article was overreaching with its inference of having demonstrated ‘trait erosion’, and that there was insufficient detail provided regarding exactly what data was being used for the analyses. To address the first of these concerns the authors have replaced any mentions of ‘trait erosion’ with terminology relating to ‘trait vulnerability’ and infer that their data shows ‘trait impairment’ following mass mortality events. To me the use of ‘trait vulnerability’ fits with the fact that authors are showing the extent to which mass mortality events impacted upon particular trait combinations. However, the use of the term ‘impairment’ still implies a temporal aspect to the data, and the monitoring of trajectories following impact, that does not exist within the data as described. All the data shows is the instantaneous impact of observed mass mortality events on different functional trait combinations.

We acknowledge that the use of “impairment” may suggest a temporal dimension. To clarify this, we have revised the manuscripts and replaced “impairment” with “impacted”, which better aligns with the nature of the data presented in this study.

Reading the manuscript it also felt that there were many instances where ‘trait erosion’ had been replaced with ‘trait vulnerability’ or ‘impairment’ without also addressing the context of the associated passage of text; meaning the passage no longer makes sense. This is particularly true of Lines 207-209 ‘Although caution is warranted, our results suggest an increase in the vulnerability of the overall trait diversity, especially if replacement by species with similar traits does not occur’.

We carefully reviewed the manuscript to ensure readability and coherence. We have rephased some sentences to clarify our findings, including the following lines:

- *“On the other hand, a prolonged high trait vulnerability may lead to an extensive loss of species with unique trait categories (e.g., tree-like species, mostly representing habitat-forming species) and may also pave the way for the establishment of new species (e.g., alien species).” [lines 237-240]*
- *“However, even if a prolonged high trait vulnerability may lead native species being replaced by alien species with similar traits, our models suggest that these species may be also vulnerable to the mortality drivers identified in this study and thus, highly threatened in the long term.” [lines 248-251]*
- *“Hence, we call for caution regarding any longitudinal gradient observed as it is challenging to unravel the cause of the lower number of mortality records and number of impacts in these areas, which could stem either from the fact that such MMEs remained unnoticed in the past or from a higher increase in trait loss from west to east.” [lines 284-287]*
- *“However, far from this ideal theoretical context, empirical evidence increasingly suggests that many tropical and temperate marine ecosystems such as coral reefs and kelp forests are experiencing decays of functionally unique or irreplaceable species (e.g., foundation species) due to an increase in the vulnerability of trait diversity which leads to shifts in functionality and loss of associated ecosystem services.” [lines 292-296]*
- *“The consistent impact on recurrent ecological trait categories not only signals a potential trend toward an increase in the vulnerability of the trait diversity over time, but also raises concern about the future potential irreversible loss of unique trait categories and associated ecosystem functions.” [lines 306-309]*

The over-inference of the results is also highlighted across Lines 440-442, in which the authors state that ‘If a single species within [a functional entity] was impaired while others [where] not, we considered the whole [functional entity] vulnerable to [the mass mortality event]’. Whilst this is a perfectly acceptable, and articulated, assumption, it does represent an over estimation of vulnerability; the implications of which are currently not discussed further within the manuscript. In summary of this point, I still feel the inferences being made by the article need toning down.

We believe that by shifting the focus from trait erosion to trait vulnerability, we are presenting our findings in a more accurate ecological terminology. Indeed, considering an entire FE as vulnerable due to the loss of a single species could lead to overestimation in the context of trait erosion. On the other hand, the concept of trait vulnerability allows identifying that the loss of one species can increase the overall risk of a decline in trait diversity. This is important because it emphasizes that even if FE remains intact, the loss of species diminishes its redundancy. To ensure caution with our findings, we have toned down key conclusions throughout this manuscript as suggested.

- *“We also reveal that 10.8% of the trait volume may have been impacted over the last five years, emphasizing the risk of a rapid ecological transformation in the Mediterranean Sea.” [lines 58-60]*
- *“Our results demonstrate that the trait volume has been impacted up to a rate of 47.4% within 35 years” [lines 148-149]*
- *“Before the year 2000, most mortality events were caused by one or two drivers, with modest effects on overall trait diversity. Impacted trait volume varied between 0 to a maximum of 10.6%” [lines 161-163]*
- *“Diseases impacted trait volumes from a maximum of 1.8% (a single FE impacted) to a maximum of 9.1% (a maximum of 9 FEs impacted)” [lines 165-167]*

- *“This upsurge, in both frequency and types of mortality drivers, led to a remarkable increase in traits impacted by mortality, with up to 26.3% of the total trait volume impacted in 2019” [lines 170-172]*
- *“This sharp increase in impacted trait volume underlines higher vulnerability in the trait diversity over time. We then averaged the impacted trait volume by decade and quantified trait vulnerability up to 0.6% in the 1990s (with an average of 3.1 ± 5.3 FEs impacted) compared to a maximum of 7.1% in the 2010s (with an average of 12.5 ± 11.3 FEs impacted).” [lines 173-176]*
- *“This decline in trait volume is even more accentuated if we average by 5 years, with a notable maximum of 10.8% impact across the Mediterranean basin in the most recent years.” [lines 177-178]*
- *“This concern of trait homogenization is emphasized by the fact that 10.8% of the trait volume may have been impacted over the past five years across the Mediterranean.” [lines 212-214]*

Finally, although the data used is now better described in the methods, this same clarity is not reflected across the rest of the manuscript. This is not a trivial issue, since in Nature Communications the methods section comes at the end of the article, and so key methodology details need to be articulated within the main text to provide context to the results. This issue is also key because the authors frequently state that the dataset that they are presenting, and using, represents a key advancement and component of the study – yet they still don’t expand on what this dataset is.

We added information regarding the dataset used in the results section as suggested.

- *“We gathered mortality data related to MMEs in the Mediterranean Sea using existing datasets from the T-MEDNet platform (<https://t-mednet.org/mass-mortality/mass-mortality-events>) and a previous study which include data from 2015 to 2019 that we complemented with a literature review (see Methods) to synthesize the most comprehensive dataset on MMEs (Fig. 1a).” [lines 107-111]*
- *“To obtain a comprehensive overview of Mediterranean benthic assemblages (including species not reported as impacted during MMEs), we used additional trait databases from previous studies assessing traits of Mediterranean benthic communities (see Methods), which resulted in a total of 389 species, subsequently classified into 228 different FEs.” [lines 139-142]*

I also detail a list of more minor concerns below:

Abstract:

Line 49: consider changing to ‘we investigated the trait patterns’.

Done.

“Utilizing extensive mass mortality events (MMEs) datasets spanning from 1986 to 2020 across the Mediterranean Sea, we investigated the trait vulnerability of benthic species that suffered from MMEs induced by nine distinct mortality drivers.” [lines 48-51]

Line 54: consider removing ‘on’.

Done.

“Our findings indicate that 55 FEs were impacted by MMEs, accentuating a heightened vulnerability within specific trait categories.” [lines 53-55]

Line 55-57: This feels like an odd example to use here, as surely the vulnerability of ‘tree-like’ morphologies centres more on the fact that Mediterranean octocoral species, which constitute the majority of tree-like morphologies, are physiological more susceptible to the mortality drivers, not that just because they are ‘tree-like’?

You are correct. However, we wanted to point out that massive morphologies were also highly impacted, which do not account only for octocorals. We have rephrased the sentence.

“Notably, more than half of the mortality records showed severe impacts on calcifying and larger species with slower growth which mostly account for tree-like and massive forms.” [lines 55-57]

Line 57-58: This statement is still inferring a temporal aspect to your data and assessment.

We rephrased it.

“Altogether, we highlight that 29 FEs suffered extreme mortality, leading to a maximum of 19.1% of the global trait volume vulnerability over 35 years.” [lines 57-58]

Line 59: Is impacted a better term than impaired? This point is relevant across the manuscript.

We changed “impaired” to “impacted” as suggested throughout the manuscript.

“We also reveal that 10.8% of the trait volume may have been impacted over the last five years, emphasizing the risk of a rapid ecological transformation in the Mediterranean Sea.” [lines 58-60]

Introduction:

Line 68: consider removing ‘the’.

Done.

“Species traits directly influence how organisms respond to environmental changes and determine their contributions to ecosystem dynamics, profoundly shaping ecosystem responses to shifts in community composition” [lines 67-69]

Line 69: responses instead of response?

Done.

“Species traits directly influence how organisms respond to environmental changes and determine their contributions to ecosystem dynamics, profoundly shaping ecosystem responses to shifts in community composition” [lines 67-69]

Line 95: What constitutes a mortality record?

Due to the journal’s word count restrictions, this information has been detailed in the methods section.

“Based on the mortality dataset 2015-2019, we considered impacted colonies or individuals displaying signs of recent mortality based on the following criteria: (1) denuded skeletons or tissue necrosis over 10% of its surface in gorgonians, sponges, and scleractinian corals and empty valves in bivalves attached to the substratum; and (2) increase of shoot mortality or sharp decline on thallus densities for seagrasses and habitat-forming macroalgal species.” [lines 349-354]

Results:

Lines 107-112: Do you mean colonies here, or individuals? Also applies to the passage of the methods section where these same criteria are outlined (Line 353).

You are right, we rephrased accordingly.

“We categorized MMEs into three severity levels: severe (mortality rate of colonies (colonial species) and individuals (non-colonial species) across sites and over time exceeding 60%), moderate (mortality rate of colonies and individuals across sites and over time between 30% and 60%), and low (mortality rate of colonies and individuals across sites and over time below 30%).” [lines 111-115]

Line 147: ‘dwindling’ is still a temporal inference.

We rephrased.

“Our results demonstrate that the trait volume has been impacted up to a rate of 47.4% within 35 years, with lower rates according to the damage severity.” [lines 148-150]

Line 168: trait instead of traits?

We believed that “traits” fits better here.

“This upsurge, in both frequency and types of mortality drivers, led to a remarkable increase in traits impacted by mortality” [lines 170-172]

Discussion:

Line 203-205: It is not clear what you are trying to say with this statement: ‘Specifically, our trait analysis identified a significantly higher vulnerability of a single trait category for each trait, surpassing other categories by 5 to 8-fold’?

This statement is referring to lines 124-126 (Figure 2). We showed that for the 10 traits accounted for, in 7 cases, a single category was highly impacted (around 80%) while the second most affected trait category was impacted less (10-15%).

“Our analysis revealed that mortality was more pronounced in specific ecological trait categories (Fig. 2), emphasizing the vulnerability of certain trait categories. Specifically, in 7 out of 10 ecological traits, a predominant trait category experienced significantly higher mortality, surpassing other categories by a minimum of 5 to 8-fold.” [lines 124-128]

Lines 230-243: This whole paragraph is discussing the implications of species loss, which is not being shown by the study.

We rephrased to introduce the implications of a prolonged high trait vulnerability.

“On the other hand, a prolonged high trait vulnerability may lead to an extensive loss of species with unique trait categories (e.g., tree-like species, mostly representing habitat-forming species) and may also pave the way for the establishment of new species (e.g., alien species). These new species may be better adapted to present and future local environmental conditions and display higher resistance and resilience to environmental perturbations due to certain traits. Currently, such species are mostly coming from the Red Sea through the Suez Canal. The disproportionate impact of invasive alien species on extant communities might result in a functional diversity shift, leading to a greater homogenization and simplification of the ecosystem. In rare cases, it might also lead to a significant increase in trait diversity. For example, the native mollusc assemblages in the eastern Mediterranean shallow benthic environments have been almost completely replaced by tropical alien species possessing distinct traits, thereby altering the overall ecosystem functioning. However, even if a prolonged high trait vulnerability may lead native species being replaced by alien species with similar traits, our models suggest that these species may be also vulnerable to the mortality drivers identified in this study and thus, highly threatened in the long term.” [lines 237-251]

Line 276: is ‘raised’ the correct verb and tense here?

We rephrased.

“Hence, we call for caution regarding any longitudinal gradient observed as it is challenging to unravel the cause of the lower number of mortality records and number of impacts in these areas” [lines 284-286]

Line 299: It is not clear what is meant by a ‘potential trend towards trait diversity vulnerability’? This is an example of when it feels like you have replaced ‘trait erosion’ with ‘trait vulnerability’ without considering the wider context of the passage of text (see main concerns).

Upon careful revision, we have ensured that the term "trait vulnerability" is used within the appropriate context throughout the manuscript. Regarding the sentence “potential trend towards trait diversity vulnerability”, we have re-evaluated the phrasing.

“The consistent impact on recurrent ecological trait categories not only signals a potential increase in the vulnerability of the overall trait diversity over time, but also raises concerns about the future potential loss of unique trait categories and associated ecosystem functions.” [lines 306-309]

Methods:

Section 4.3 – Is there a clearer way to distinguish the different listing notation used across categories within traits, and the different traits used? Currently, the traits a listed using capitalised letters, whilst the within trait categories are listed using either lower case lettering or numbering. Overall, this makes the paragraph and listing difficult to follow.

We aimed to distinguish between qualitative traits (indicated by lowercase letters) and semi-quantitative traits (indicated by numbers), ensuring consistency with previous foundational studies for

this database (Teixidó et al. 2021, 2024; Gómez-Gras et al. 2021; Galobart et al. 2023). We have revised this section accordingly, incorporating this information. We also recommend that readers refer to Table S3.

“The traits were categorized as qualitative (defined for each following category with a lowercase letter) and semi-quantitative (defined for each following category with a number)” [lines 380-382]

Reviewer #1 (Remarks on code availability):

There is no code repository at the link provided.

The repository was private. It is now public in the revised version.

Reviewer #2 (Remarks to the Author):

This is the second time I have seen the manuscript. The authors have made generally satisfactory responses to my concerns, and have improved clarity in general.

We thank Reviewer 2 for this positive feedback.

I do have a few remaining substantial concerns:

* Data availability: the authors wrote in their response they would provide machine-readable open data. I didn't see it on the journal website yet; there is Table S5 in 527652_0_supp_9383202_sh84yj.pdf but this is not in CSV or similar format, and it is unclear if this data alone can be used to reconstruct the full study

The repository was private. It is now public.

The data can be found here: https://github.com/JayCrlt/MMEs_Mortality/tree/master/Data/R

* Code availability: the provided github: https://github.com/JayCrlt/MMEs_Mortality does not work when I tried to load it. I was unable to verify the code base.

The repository was private. It is now public.

Also a few minor concerns:

* many of the selected verbs still feel to me inappropriate in tone (e.g. 'unveiling', 'accentuating', 'weakening'). The authors may want to consider changing some of these, but it is really a stylistic choice.

We carefully reviewed the manuscript to ensure readability and have toned down the manuscript, as also suggested by reviewer 1.

* Fig 4: the legend is still not clear - do the colors represent the combined predicted effects of the listed variables, from a model? If so, describe the model in a bit more detail in the caption.

We added this information to the caption. It does not result from the model, but rather from subsetting the data accordingly and calculating the volume for each mortality driver over time.

“Fig. 4 | Impact of global and local mortality drivers on trait volume across the Mediterranean Sea from 1986 to 2020. The heatmap represents the volume of the trait space impacted due to mortality drivers over time. a, Abiotic mortality drivers are: temperature anomaly, storm, pollution, increase of turbidity, others, and their combination; b, Biotic mortality drivers are: disease, mucilage coverage, predator outbreak, others, and their combination; c, Combination of both biotic and abiotic mortality drivers on the ecological trait volume. The trait volume space has been calculated for each mortality driver as well as for overall mortality drivers over time and colour-coded accordingly.” [lines 541-547]

Reviewer #2 (Remarks on code availability):

See above.

The repository was private. It is now public.

Reviewer #3 (Remarks to the Author):

While the ms tries to address many of my points raised, it has not eliminated my concerns that it lacks scientific rigour. One important point remains, the potential collinearity between the sampling effort and

the number of MMEs through time and external drivers (see below). The other the selection of traits and subset of fauna sampled which will drive the analysis towards specific traits sets to be identified as vulnerable (i.e. sessile large species that can easily be observed vs mobile species). And last the reduction of trait volume is a somewhat abstract measure, that appears attractive when telling the story, but is difficult to translate into real world scenarios. Overall I feel the paper is aiming very high, decorated with nice looking figures, but lacking rigour and substance. I do think its worthwhile publishing but not without an in depth exploration of its limitations (and as such I am not sure this is the right paper for nature communications).

- **1) Regarding the collinearity between sampling effort and the number of MMEs over time, we acknowledge that this is a critical point to address. Therefore, we reanalyzed the data using 100 observations per decade to highlight the increase in the vulnerability of trait volume over time. This is now presented as a supplementary figure (Figure S4) and used in the results to validate our previous findings (see the next response for a more in-detailed explanation).**

- "To ensure that our findings were not biased due to an unbalanced sampling effort, we randomly picked 100 observations each consecutive decade and defined the corresponding trait volume, repeating this process 1,000 times, and validating the escalating vulnerability observed in trait volume over time." [lines 178-182]

- **2) Regarding the selection of traits, you are correct that some mortality events are easier to assess than others. However, this study specifically addresses MMEs, which have been thoroughly investigated by experts aiming to quantify such events across various taxa. MMEs involve assessing the health status of benthic communities by quantifying the percentage of mortality across individuals and colonies during field surveys. Importantly, these field surveys did not focus solely on large species but rather on a broad spectrum of benthic species, including small, motile, and solitary organisms. While your concern about potential bias is valid, the overrepresentation of specific trait categories is also consistent with ecological dynamics. Motile species are more likely to evade mortality drivers, while sessile organisms are not. Additionally, as Table S3 shows, categories 3 (medium) and 4 (large) include organisms ranging from 5 to 50 cm, making them easier to assess. However, most of the mortality records were for larger individuals.**

- "The T-MEDNet dataset contains mortality data assessed and validated exclusively by scientific experts, accounting for 710 records across a wide spectrum of benthic species. The mortality dataset 2015-2019 was obtained through a collaborative effort across the Mediterranean Sea and represents the most comprehensive inventory of MME records for benthic species in the region from 2015 to 2019, accounting for 1,125 records." [lines 324-328]

- **3) Regarding the use of trait volume, we suggest thinking of it as an index. Specifically, the higher the trait volume, the more vulnerable the ecosystem. This measure has been widely used in prestigious journals over the past decade (see references below). Additionally, we have included in our manuscript the number of affected functional entities, which can help to provide a global picture and are highly relevant in the context of global change, as the loss of a single species can be compensated by another species with similar traits within the ecosystem.**

References

- Carmona et al. (2021). Erosion of global functional diversity across the tree of life. *Science Advances*.

- *Toussaint et al. (2021). Extinction of threatened vertebrates will lead to idiosyncratic changes in functional diversity across the world. Nature communications.*
- *McWilliam et al. (2020). Deficits in functional trait diversity following recovery on coral reefs. Proceedings of the Royal Society B.*

- **4) Finally, we believe that this study is crucial in the context of global change, which is why we aimed high. With the inclusion of the new analysis presented in Figure S4 and our efforts to address the collinearity between time and sampling effort, we are confident in our conclusions regarding the increasing vulnerability of trait volume in benthic communities over time.**

I am still sceptical about some of the presented results especially with respect to the temporal trends. I think the authors should provide more details about the sampling effort through time. MMEs observation and sampling effort may have increased simultaneously with time, potentially confounding the data presented and the conclusions drawn. The authors should be transparent about their sampling effort and should provide measures to adjust for this imbalance in sampling, in their analysis and interpretation of the data. In the response to the reviewers comments this point has not yet been properly addressed. “We also have to notice that although a consistent sampling effort over space and time is the ideal situation, it is impossible to be obtained on a large scale such as the entire Mediterranean basin involving 22 coastal countries and spanning 35 years. Still, the use of the best available datasets over large spatial and temporal scales can provide extremely useful insights and is a common practice in global or large regional assessments in ecological studies”. At least this point needs to be acknowledged and the limitations with respect to drawing firm conclusions need to be modulated.

To ensure transparency, the data, code and supplementary materials were designed to be publicly available in the original submission. Following the comments of Referee #3 regarding the need to provide more details about the sampling effort through time, we have added a new supplementary figure (Figure S4) to show the number of studies included, the number of mortality records considered, and the affected volume per decade, based on a uniform sampling effort (*i.e.*, random sampling of 100 mortality records per decade), in order to support our narrative and conclusions. We acknowledge that potential collinearity between the sampling effort and the number of MMEs through time and external drivers was a critical issue to address, and we believe that by adding this new analysis, we have responded to the main concern regarding this study.

“To ensure that our findings were not biased due to an unbalanced sampling effort, we randomly picked 100 observations each consecutive decade and defined the corresponding trait volume, repeating this process 1,000 times, and validating the escalating vulnerability observed in trait volume over time.” [lines 178-182]

“Figure S4 | Overview of the sampling effort and related measures. a. Number of studies assessing at least one MME over time, compiled in the Mass Mortality Event dataset. b. Number of mortality records assessed in the Mass Mortality Event dataset over time. c. Increase in the vulnerability of trait volume over time, based on random sampling of 100 mortality records per decade. This process has been repeated 1,000 times to strengthen our conclusions, with each iteration represented by a grey dot. The average value is indicated by a red dot. A logarithmic model fit is included to illustrate the increase in trait volume vulnerability over time, yielding an R^2 of 0.99” [lines 757-763 – supplementary information]

Additionally, to ensure caution with our findings, we toned down key conclusions in the manuscript.

- *“We also reveal that 10.8% of the trait volume may have been impacted over the last five years, emphasizing the risk of a rapid ecological transformation in the Mediterranean Sea.” [lines 58-60]*
- *“Our results demonstrate that the trait volume has been impacted up to a rate of 47.4% within 35 years” [lines 148-149]*
- *“Before the year 2000, most mortality events were caused by one or two drivers, with modest effects on overall trait diversity. Impacted trait volume varied between 0 to a maximum of 10.6%” [lines 161-163]*
- *“Diseases impacted trait volumes from a maximum of 1.8% (a single FE impacted) to a maximum of 9.1% (a maximum of 9 FEs impacted)” [lines 165-167]*
- *“This upsurge, in both frequency and types of mortality drivers, led to a remarkable increase in traits impacted by mortality, with up to 26.3% of the total trait volume impacted in 2019” [lines 170-172]*
- *“This sharp increase in impacted trait volume underlines higher vulnerability in the trait diversity over time. We then averaged the impacted trait volume by decade and quantified trait vulnerability up to 0.6% in the 1990s (with an average of 3.1 ± 5.3 FEs impacted) compared to a maximum of 7.1% in the 2010s (with an average of 12.5 ± 11.3 FEs impacted).” [lines 173-176]*
- *“This decline in trait volume is even more accentuated if we average by 5 years, with a notable maximum of 10.8% impact across the Mediterranean basin in the most recent years.” [lines 177-178]*
- *“This concern of trait homogenization is emphasized by the fact that 10.8% of the trait volume may have been impacted over the past five years across the Mediterranean.” [lines 212-214]*

Finally, we believe that this study presents the most up-to-date dataset on Mediterranean MMEs and provides a novel analysis of its implications, making it highly relevant for assessing the impacts of global change on benthic communities.

The title should contain also the word benthic I suggest the following title: Trait diversity vulnerability across the Mediterranean Sea following Mass Mortality Events"

We appreciate your suggestion. We have made the change accordingly.

“Vulnerability of benthic trait diversity across the Mediterranean Sea following Mass Mortality Events” [lines 1-2]

REVIEWERS' COMMENTS

Reviewer #1 (Remarks to the Author):

This is the third time I have seen the manuscript. The authors have addressed my concerns about data availability. I successfully viewed the repository and confirmed the data / scripts in several files.

I thought the other reviewers raised important points and feel satisfied by the ways the authors have addressed them.

We thank reviewer 1 for these positive comments and constructive feedbacks over the 3 rounds of reviews.

I have just a few last language suggestions:

abstract: "we highlight that 29 FEs suffered extreme mortality, leading to a maximum of 19.1% of the global trait volume vulnerability over 35 years." - grammar is not clear to me, is this 'maximum loss'? something is missing here. also, 'may have been impacted' - how has it been impacted? 'temporarily lost'?

We have rephrased both sentences.

- *"Altogether, we highlight that 29 FEs suffered extreme mortality, leading to a maximum increase of 19.1% of the global trait volume vulnerability over 35 years." (lines 57-59)*
- *"We also reveal that 10.8% of the trait volume may have been temporarily lost over the last five years, emphasizing the risk of a rapid ecological transformation in the Mediterranean Sea." (lines 59-61)*

Reviewer #1 (Remarks on code availability):

This looks good to go now.

Reviewer #3 (Remarks to the Author):

This is the third opportunity I have had to review this piece of work, and I feel that Carlot et al. have appropriately addressed my concerns. This piece of research comprises a huge amount of effort in synthesising an extensive database on the severity of mass mortality events across the Mediterranean and their impact upon functional trait entities. The manuscript now outlines the work with sufficient clarity as for a broad audience to follow and encompasses an insightful and thought-provoking discussion of the results. Thank you for the opportunity to review this piece of work.

As reviewer 1, we thank reviewer 3 for these positive comments and constructive feedbacks over the 3 rounds of reviews.

I include below a few minor comments:

Line 54: consider changing to: 'indicate that 55 **of these** FE's were impacted by'.

Done.

"Our findings indicate that of these 55 FEs were impacted by MMEs, accentuating a heightened vulnerability within specific trait categories." (lines 53-55)

Line 58: should this line be: 'leading to a maximum **increase** of 19.1%'?

Done.

"Altogether, we highlight that 29 FEs suffered extreme mortality, leading to a maximum increase of 19.1% of the global trait volume vulnerability over 35 years." (lines 57-59)

Line 149 – 150: can you clarify what you mean exactly by 'with lower rates according to the damage severity'?

We rephrased.

"Our results demonstrate that the trait volume has been impacted up to a rate of 47.4% within 35 years, with lower rates associated with high damage severity." (lines 149-151)

Line 180: consider changing to: 'picked 100 observations **from** each consecutive decade'.

Done.

“we randomly picked 100 observations from each consecutive decade” (lines 180-181)

Line 205: consider changing to: ‘and **their associated** species-specific traits’.

Done.

“By compiling one of the most comprehensive marine temperate mortality databases and their associated species-specific traits” (lines 205-206)

Line 222: consider changing to: ‘This vulnerability **in** calcifying and habitat-forming organisms’.

Done.

“The pointed out vulnerability in calcifying and habitat-forming organisms” (line 223)

Line 248: consider removing the word ‘even’.

Done.

“However, if a prolonged high trait vulnerability may lead native species being replaced by alien species with similar traits” (lines 249-251)

Line 321: The citation for the T-MEDNet platform should appear when this database is first mentioned on line 108.

Done.

“Mortality data have been acquired from three main sources: (1) from the T-MEDNet platform collecting data since 2012 (<https://t-mednet.org/mass-mortality/mass-mortality-events>)⁵¹”

Reviewer #3 (Remarks on code availability):

Exceptionally clear landing page to walk the readers through reproducing the analysis and figures. Nice use of a run-script.

Thank you!